# mREDDITSUM: A Multimodal Abstractive Summarization Dataset of Reddit Threads with Images

**Keighley Overbay[1]    Jaewoo Ahn[1*]    Fatemeh Pesaran zadeh[1*]**
**Joonsuk Park[2,3,4]    Gunhee Kim[1]**

[1]Seoul National University    [2]University of Richmond    [3]NAVER AI Lab    [4]NAVER Cloud

{keighley.overbay,jaewoo.ahn,fatemeh.pesaran}@vision.snu.ac.kr

park@joonsuk.org, gunhee@snu.ac.kr

http://vision.snu.ac.kr/projects/mredditsum

## Abstract

The growing number of multimodal online discussions necessitates automatic summarization to save time and reduce content overload. However, existing summarization datasets are not suitable for this purpose, as they either do not cover discussions, multiple modalities, or both. To this end, we present mREDDITSUM, the first multimodal discussion summarization dataset. It consists of 3,033 discussion threads where a post solicits advice regarding an issue described with an image and text, and respective comments express diverse opinions. We annotate each thread with a human-written summary that captures both the essential information from the text, as well as the details available only in the image. Experiments show that popular summarization models—GPT-3.5, BART, and T5—consistently improve in performance when visual information is incorporated. We also introduce a novel method, cluster-based multi-stage summarization, that outperforms existing baselines and serves as a competitive baseline for future work.

| Post: | Post Summary: |
|---|---|
| 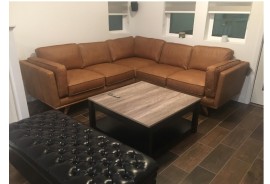 We got this couch for our living room and I need help finding the perfect coffee table to go with it. ... | The OP asked for help with finding the right coffee table shape to match their brown leather sectional. |
| **Comments:**
**User 1:** I would do a circular table.
**User 2:** Definitely round! There are a lot of sharp angles already ...
**User 3:** There's way too much furniture in this space, the ottoman has to go ...
**User 4:** I think you should look for a natural wood triangular shaped coffee table ...
**User 5:** You should get a rug. ... | **Comment Summaries:**
**C1,C2,C4:** Commenters suggest a differently shaped coffee table from the square one in the picture, round or triangular or hexagonal.
**C3:** A commenter suggests eliminating the ottoman as it takes up too much space.
**C5:** A commenter suggests adding a rug. |
| **Full Summary:**
The OP asked for help with finding the right coffee table shape to match their brown leather sectional. Commenters suggested a differently shaped coffee table from the square one he has already, such as round, triangular, or hexagonal. A few commenters suggested eliminating the ottoman, as it is too big for the small space. Others suggested adding a rug. | |

Table 1: An example from the mREDDITSUM dataset. Both the post, several viewpoints from the comments, and the overall thread are summarized along with important content only available in the image (in green), or in both image and text (in blue).

## 1 Introduction

With the increased popularity of online discussion forums like *Reddit*, discussion threads that consist of a post and various numbers of comments, have quickly accumulated. It thus becomes overwhelming for users to sift through the threads to find the information they seek, which in turn has led to the development of automated means for text-only discussion summarization (Bhatia et al., 2014; Fabbri et al., 2021, 2022).

However, discussion threads are often multimodal, containing visuals in addition to text. This added modality cannot be ignored, as it plays a key role in the respective discussions. For example, in Table 1, the image of the couch is essential for discussing which coffee table would go well with it.

Yet, multimodal summarization has so far been limited to news and instructional domains (Zhu et al., 2018; Sanabria et al., 2018; Palaskar et al., 2019; Liu et al., 2020) that are not easily transferable to online discussions surrounding images.

To fill the gap, we tackle multimodal discussion summarization. In particular, we consider Reddit discussion threads in which the post solicits advice regarding an issue described with an image and text, and commenters offer opinions, as opposed to simple reactions or jokes. Here, the goal is to generate an abstractive summary faithfully capturing the information from the post—both image and text—and comments. This task is especially challenging because along with the need to effectively process the multimodal input, a quality summary

---

[*]Equal contribution.

must provide good coverage of commenters' varying perspectives and opinions without redundancy.

To facilitate research in this direction, we contribute the Multimodal Reddit Summarization (MREDDITSUM) dataset, consisting of 3,033 Reddit discussion threads containing posts (text and image) and comments (text-only), each accompanied by a human-written summary, as shown in Table 1. We carefully select subreddits with discussions surrounding an image, and collect summaries that not only summarize the text, but also make reference to relevant information present only in the image. See Appendix C for more examples.

We also propose cluster-based multi-stage summarization (CMS), a novel method to summarize multimodal discussions. It processes discussions in three stages: (i) comments are first clustered by similarity, (ii) each cluster is summarized in a sentence, and (iii) the cluster-summaries are summarized. Experiments show that CMS consistently outperforms popular large language models (LLMs) for summarization—GPT-3.5 (Brown et al., 2020), BART (Lewis et al., 2020), and T5 (Raffel et al., 2020). Also, incorporating image information, either as a dense vector or in text caption, consistently boosts the performance.

Our main contributions are as follows:

- We present MREDDITSUM, the first multimodal discussion summarization dataset with human-written summaries with essential information from both the text and the image. Table 2 presents comprehensive comparison with existing summarization datasets.

- We propose cluster-based multi-stage summarization (CMS), a novel method to summarize multimodal discussions outperforming competitive baselines like GPT-3.5, BART and T5, as well as their vision-guided variations.

## 2 Related Work

**Discussion Thread Summarization.** Despite the prevalence of discussion threads online, it has been understudied for summarization. This is likely due to the fact that it can be a difficult target for extractive summarization, though there has been small extractive summarization dataset (Bhatia et al., 2014).

More recently, several abstractive summarization datasets have been proposed. ConvoSumm (Fabbri et al., 2021) presented a dataset of 2000 summarized forum threads, 500 from each of 4 different domains including NYT articles, Reddit, StackExchange, and Email threads. Answer-Summ (Fabbri et al., 2022) is another dataset consisting of 4,631 question-answering discussion threads sourced from StackExchange. AnswerSumm shares the most similarities with our dataset, as they also summarize multi-speaker threads, and their annotation pipeline shares some similarities with ours. They also cluster the comments and summarize these groups before going through a final summary editing process, similar to our pipeline. The key differences between this dataset and ours is that AnswerSumm is only text-based with no images and operates in a different domain, as they are all question-answering threads from StackExchange. In contrast, our dataset includes both images and text, and focuses on Reddit threads where the images play a key role. Additionally, in our annotation pipeline we summarize the original post and image as well, which to our knowledge has not been done in any other forum summarization dataset. This is useful because often the posts alone have unclear intent that may require context derived from the image or forum domain itself.

Other related summarization datasets include multi-turn datasets such as SamSUM (Gliwa et al., 2019), consisting of chat-like dialogues and human-annotated summaries, and EmailSum (Zhang et al., 2021), containing work-related emails and both long and short reference summaries.

Overall, though there is a small variety of existing discussion thread summarization datasets, they are all currently only text-based and none of these tackle both original post and thread summarization.

**Multimodal Summarization.** Though other multimodal research areas such as VQA (Agrawal et al., 2017) and text-image pretraining (Radford et al., 2021; Li et al., 2022a, 2023) have been gaining attention in recent years, there only exist a small handful of works that address multimodal summarization, which aims to generate a summary that includes salient information from inputs with multiple modalities. For example, MSMO (Zhu et al., 2018; Qiu et al., 2023), Multimodal Summarization with Multimodal Outputs, takes inputs of various modalities and outputs both a text-based summary and a representative image.

However, our task aims to generate a unimodal output—that is, a purely textual summary. This is similar to the multimodal summarization done on the How2 Dataset (Sanabria et al., 2018; Palaskar

| Dataset | Domain | # Docs | Doc Len | Sum Len | # Turns | # Speakers | Modality | StructSum |
|---------|--------|--------|---------|---------|---------|------------|----------|-----------|
| MREDDITSUM(ours) | Forum | 3,033 | 691.0 | **91.0** | **22.6** | **15.59** | *t, i* | ✓ |
| AnswerSumm | Forum | 4,631 | 787.0 | 47.0 | 6.4 | 6.17 | *t* | ✓ |
| ConvoSumm$_{reddit}$ | Forum | 500 | 641.0 | 65.0 | 7.88 | * | *t* | |
| SamSUM | Dialog | 16,396 | 124.1 | 23.4 | 12.19 | 2.39 | *t* | |
| CNN/DM | News | 286,817 | 766.0 | 53.0 | 1 | 1 | *t* | |
| MSMO DailyMail | News | 314,581 | 722.7 | 55.0 | 1 | 1 | *t, i* | |
| How2 | Video | 79,114 | 291.0 | 33.0 | 1 | 1 | *t, v* | |

*: speaker info not provided / t: text / i: image / v: video*

Table 2: A comparison of MREDDITSUM and other summarization datasets. Among forum-based and multi-turn datasets, MREDDITSUM is the only multimodal dataset, and it has the highest summary length, number of turns, and number of speakers. Length is reported in the average number of words, and turns are the average number of each instance of a post, comment, or speaker change. StructSum denotes whether there are structural-level summary annotations, such as for comment clusters. Statistics are taken from the respective papers for AnswerSumm (Fabbri et al., 2022), ConvoSumm(Fabbri et al., 2021), SamSUM (Gliwa et al., 2019), CNN/DM (Nallapati et al., 2016), MSMO DailyMail (Zhu et al., 2018), and How2 (Yu et al., 2021).

et al., 2019), where a textual transcript of the video along with the video frames are generated into a text summary. (Yu et al., 2021) reported that incorporating the additional modality of the video frames into their summarization models showed improvement compared to text-only based models. Though this multimodal summarization task is the most similar to ours, there are some key differences. The How2 dataset uses short video captions as pseudo-summaries, instead of detailed human-annotated summaries like we curate for MREDDIT-SUM. Additionally, our text is a rich multi-speaker discussion, rather than a transcript of audio. Finally, MREDDITSUM's threads are specifically selected to include images where their information is necessarily included in the summary, whereas there is no such assurance for How2's videos.

## 3 The MREDDITSUM Dataset

### 3.1 Data Preparation

To construct a meaningful multimodal discussion summarization dataset, we imposed three major criteria when selecting Reddit threads.

**Criterion 1.** The discussion thread needs to contain an image. Since Reddit does not allow images embedded in comments, this means that the post needs to contain an image.

**Criterion 2.** The discussion needs to be centered around an image in such a way that the information only available from the image plays a key role in the discussion. In some threads, the image does not provide any information, e.g. it is a favorite character of the original poster. In such cases, simply

summarizing the text is sufficient, and a multimodal model is unnecessary.

**Criterion 3.** The discussion needs to be meaningfully summarizable. Many Reddit threads that include images are meant to incite reactions from other users, or to be shared in a jocular manner. On the other hand, some threads consist of the post clearly asking for advice or opinions, thereby eliciting diverse responses from a number of commenters. Summarizing these opinions along with the advice would be helpful for readers to understand the gist of the threads.

**Sources.** Given the aforementioned criteria, we identify 9 subreddits, e.g. r/fashionadvice (See Appendix A for the complete list), which consist primarily of image-based posts where the original poster is soliciting advice or opinions about either clothing or interior design. We collect all threads from these subreddits with over 5 comments from years 2015-2022. Collection is done with RedCaps (Desai et al., 2021) modified to collect all comments from each thread. We additionally follow similar preprocessing steps (Ahn et al., 2023), removing all posts that contain NSFW content or images with faces. We also remove any comments with NSFW content, and comments posted by bots. All responses to these removed comments are also removed. We replace URL's with the "[URL]" token.

### 3.2 Annotation

We then annotate the data after selecting qualified workers from Amazon Mechanical Turk (MTurk). We limit our workers to those from English-

speaking countries with a HIT approval rate over 98%, with greater than 5000 HITs approved. For all tasks, workers are required to pass a qualification task where the results are manually checked for quality. Any workers who are found to submit low-quality work have their qualification revoked. In total, a pool of 40 annotators were qualified to perform annotation tasks. As each task was performed separately, it was possible for multiple annotators to contribute to a single summary. so Additional detail on the annotation interface and instructions are found in Appendix B. The annotation is conducted in a 3-step annotation pipeline as follows.

**Step 1: Original Post Summarization.** In the first step, we present workers with the original post along with the image from that post. We ask the annotators to summarize in a single sentence the intent of the original poster, as well as the most relevant details from the image. We use this method because a post that simply reads "blue or black?" may only be comprehensible when paired with the image of blue and black heels next to a blue dress, and a true text-only summary should be comprehensible without the image. Our summary may then read "The original poster asked if blue or black heels would match better with a strapless, knee-length blue dress," eliminating the need of the image to comprehend the question. In this way, all information necessary to understand the question should be self-contained within the summary.

**Step 2: Comment Cluster Summarization.** We first cluster the comments to identify groups of comments that share a similar opinion. We follow the method described in AnswerSUMM (Fabbri et al., 2022) to allow for clusters of varying sizes and number. We use a RoBERTa-based model fine-tuned for semantic similarity to get sentence embeddings of the top-level comments from each thread. We then use agglomerative clustering with average linkage, cosine distance, and a maximum distance of 0.5 to generate clusters of comments.

We then rank the comment clusters according to their size and Reddit score. We take the sum of all Reddit scores of the top-level comments in a single cluster as a saliency score of the cluster. We select the top 5 clusters with the highest saliency scores and use these for annotation. We do this to limit the size of the summary and to help remove irrelevant comments, while encouraging larger clusters of comments with a similar sentiment.

We then present these groups of comments to an-

| Structure | Document | Summary |
|---|---|---|
| Original Post | 1.62 sents | 1.07 sents |
| | 18.87 words | 23.14 words |
| Comment | 6.63 sents | 1.34 sents |
| Clusters | 85.05 words | 20.17 words |
| | 21.6 comments | |
| Full Thread | 37.41 sents | 5.32 sents |
| | 691 words | 91.0 words |

Table 3: Average statistics across the original post, comment clusters, and full thread structures of our dataset.

notators along with the original post and image, and ask them to summarize within one or two sentences the main opinions present in each group of comments. We encourage the annotators to reference objects or details from the image when necessary. For consistency, we instruct the annotators to refer to the commenters as "Commenters" as opposed to people, users, or other words.

**Step 3: Summary Synthesis.** Finally, we concatenate the original post summary as well as the comment cluster summaries, in descending order of their saliency-scores. We then present these summaries once more to annotators and ask them to edit them for fluency and readability. We encourage annotators to reduce repetitive wording, add connectives between sentences, and to rearrange sentences so that related topics are next to each other and the overall summary reads as more natural. We also ensure all summaries are written entirely in the past-tense.

### 3.3 Dataset Analyses

**Statistics**. The resulting dataset contains a total of 3,033 posts and summaries. We split these into a train, test, and validation set of sizes 2729, 152, and 152, respectively. We present further statistics in Table 2, where we compare with similar summarization datasets from a few different domains. The average summary length for MREDDITSUM is longer than other datasets; however, this is not surprising given the nature of summarizing varying opinions, of which there could be many. Additionally, we describe the structure-level statistics in Table 3; note that while the average length of the Original Post summary is longer than the document, this is due to the additional image description and context. For the full thread, the summary is 13.2% as long as the input on average, which is comparable to SamSUM's 19% and How2's 11.3%.

**Summary Quality**. In order to ensure the quality summary, we additionally performed an exper-

iment to rate the annotated summaries out of 3 metrics. Following a closely related work, Sam-SUM (Gliwa et al., 2019), we randomly selected 100 thread-summary pairs and had 2 independent judges from MTurk grade them on a quality scale of -1, 0, or 1, where -1 means a summary is irrelevant or does not make sense, 0 means a summary extracts only part of the relevant information or makes some mistakes, and 1 means a summary that is understandable and delivers a brief overview of the relevant information from the thread. We asked annotators to score summaries on overall quality, fluency, and faithfulness, similar to our human annotation found in Sec 5.2. We found the scores were highly positive, with average scores of 0.95, 0.96, 0.83 for overall quality, fluency, and faithfulness respectively. Additionally, we found Gwet's AC1 agreement scores of .91, 0.89, and 0.53, corresponding to high, high, and moderate agreement, respectively. Note that we used the Gwet's AC1 score for interannotator agreement as it performs well despite class imbalances where agreement is high. (Gwet, 2008; Wongpakaran et al., 2013; Wong et al., 2021)

**Abstractiveness**. Extractive-Oracle ROUGE scores in Table 4 show that our dataset is similar in abstractiveness to other multi-turn datasets, and much more abstractive than DailyMail. Though scores are not available for MSMO, it is expected that the scores would be similar to DailyMail.

**Relatedness between Text and Images**. We also calculate the CLIPScore (Hessel et al., 2021), a metric that measures the correlation between text and an image, to determine how grounded our summaries are to the images from each thread. Our summaries have an average CLIPScore of 74.62, the post summaries alone achieve 74.89, and the comment cluster summaries alone score 68.34. These suggest our summaries, especially the post summaries, are well-correlated with the images.

## 4 Experiments

### 4.1 Task Description

We consider the multimodal summarization task, where the input includes all original text and the image and the output is a text-only summary that describes both the document and image. The text includes the post and comments, and the goal is to accurately summarize both the original poster's intent and commenters' opinions. For this task, we format the text input as the following: "Orig-

| Dataset | Extractive Oracle ROUGE | | |
|---|---|---|---|
| | R1 | R2 | RL |
| MREDDITSUM (ours) | 36.52 | 11.95 | 31.42 |
| AnswerSumm | 40.05 | 18.45 | 35.70 |
| ConvoSumm$_{reddit}$ | 35.74 | 10.45 | 30.74 |
| DailyMail | 55.23 | 30.55 | 51.24 |

Table 4: A comparison of Extractive Oracle ROUGE scores on MREDDITSUM and related datasets. The lower the score, the more abstractive the summaries are. Results for related works are from the respective papers (Fabbri et al., 2022, 2021).

inal Post: Original Post", with "Image: Image Caption." appended for models that include image captions. We then additionally append the comments in the form "User 1: Comment 1. User 2: Comment 2. ...", where each username has been anonymized. Comments are listed in the order that they are scraped from Reddit in. The target output is the result of our final summary.

### 4.2 Evaluation Metrics

Following the standard metric for summarization evaluation, we use the ROUGE (Lin, 2004) and BertScore (Zhang* et al., 2020). ROUGE[1] measures the salience of model-generated summaries by comparing n-grams in the generated summary and gold summary. We consider three variants: ROUGE-1/2 (R1/2) measure the unigram/bigram overlap, and ROUGE-L (RL) determines the longest common subsequence between summaries. BertScore[2] computes a soft token similarity using contextual embeddings output from BERT, instead of word matches. We use the default RoBERTa-large model and rescale with baseline.

### 4.3 Baseline Models

We consider four text-only baseline models: GPT-3.5 (zero-shot), BART, T5, and LongT5 (fine-tuned), as well as their extensions to make use of image information, either as image captions or embeddings.

**Extractive Baselines (Lead-1, Lead-Comment, Ext-Oracle)**. Lead-1 uses the first sentence from the document as the summary, and Lead-Comment uses the leading top 5 comments from the thread. Ext-Oracle extracts passages from the document to achieve the maximum possible ROUGE score, and

---

[1]https://github.com/pltrdy/rouge
[2]https://github.com/Tiiiger/bert_score

thus is the highest possible performance from an extractive model.

**Text-only Baselines (GPT-3.5, BART, T5, LongT5).** GPT-3.5 (Ouyang et al., 2022) is an LLM that has shown excellent zero-shot performance in summarization tasks (Goyal et al., 2022; Bhaskar et al., 2023). We use the largest model, text-davinci-003, through the OpenAI API, with the prompt "Summarize what the original post was asking about and the general opinions of the commenters.", which is determined empirically to perform well and closely mimic the instructions given to annotators. We also evaluate three fine-tuned models, BART-base (Lewis et al., 2020), and T5-base (Raffel et al., 2020), which are high-performing LLMs with good summarization abilities, as well as LongT5-base (Guo et al., 2022) which is an extension of T5 that is capable of handling longer input sequences. We pretrain them on the CNN/DailyMail (Nallapati et al., 2016) summarization dataset before fine-tuning it for our task.

**Extensions with Image Captioning (GPT3.5-ImgCap, BART-ImgCap, T5-ImgCap, LongT5-ImgCap).** We extend the text-only baselines to incorporate visual information through the use of an image caption, denoted as GPT3.5-ImgCap, BART-ImgCap, T5-ImgCap, and LongT5-ImgCap respectively. They take advantage of powerful LLMs without large amounts of multimodal training to understand visual features. For image captions, we use the BLIP2 model (Li et al., 2023) trained on COCO image captions (Chen et al., 2015) and generate multiple image captions for each image using nucleus sampling. Since a more detailed and grounded image caption that describes concrete objects is best for this task, we use a image-grounding model, GLIP (Li et al., 2022b), to score each caption by grounding it with the image, and calculate how many image-text grounded pairs are above a threshold of 0.7. We then select the image caption with the highest score and append the caption to the input after the original post. We then fine-tune BART-ImgCap, T5-ImgCap, and LongT5-ImgCap as described above; for GPT3.5-ImgCap, we use the caption-appended prompt.

**Extensions with Vision-Guidance (VG-BART, VG-T5).** Vision-Guided BART and T5 are presented in (Yu et al., 2021) for multimodal summarization. They include additional visual layers that receive video embeddings as input, and show state-of-the-art performance in multimodal summariza-

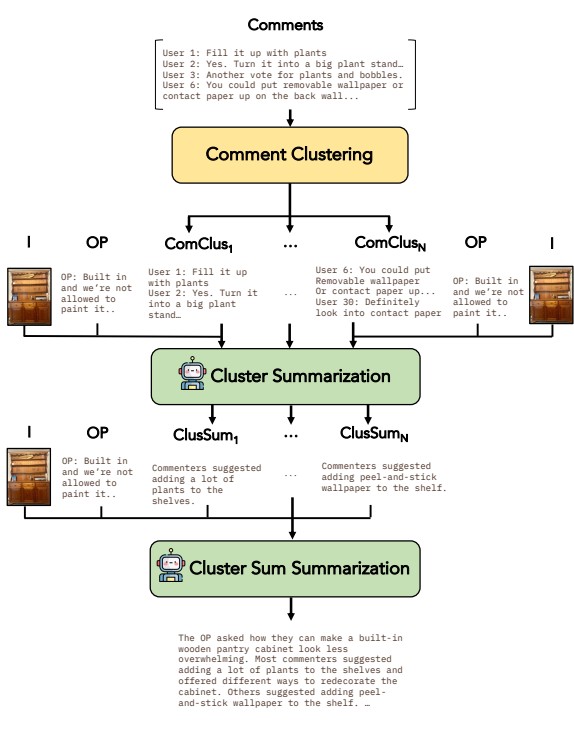

Figure 1: An illustration of Cluster-based Multi-stage Summarization (CMS): (1) comments are first clustered by similarity, (2) each cluster is summarized in a sentence, and (3) the cluster-summaries are summarized. (OP: the original post / I: the post image / $ComClus_k$: the $k$-th comment cluster / N: the number of comment clusters / $ClusSum_k$: the generated summary of the $k$-th comment cluster)

tion for the How2 dataset. We modify the original models by instead using 768-D ViT-base (Dosovitskiy et al., 2021) image embeddings as input, as they have shown excellent performance as an image backbone. We use cross-modal dot product attention with a forget-gate and image transformer, as this version performed best in our experiments. We use the same T5-base and BART-base pretrained on CNN/DM to initialize the encoder and decoder. For VG-BART, we pretrain the visual layers using the COCO image captions before fine-tuning on our dataset; VG-T5 shows no performance increase from visual pretraining, so we initialize its layers from scratch.

## 4.4 Cluster-based Multi-stage Summarization

One challenge in summarizing discussions is that they can be very long. To confirm that this is causing an issue, we conduct a preliminary experiment on the fine-tuned BART model by comparing the results of two different test subsets: the *long* subset with more than 22 turns and the *short* subset with

| Model | R1 | R2 | RL | BertS |
|---|---|---|---|---|
| *Extractive* | | | | |
| Lead-1 | 15.23 | 3.46 | 13.24 | 11.89 |
| Lead-Comment | 22.86 | 5.55 | 20.43 | 7.16 |
| Ext-Oracle | 36.52 | 11.95 | 31.42 | 16.71 |
| *Zero-shot Prompting* | | | | |
| GPT-3.5 | 34.29 | 9.10 | 30.39 | 30.15 |
| GPT-3.5-ImgCap | **34.59** | **9.41** | **30.59** | **31.07** |
| *Fine-tuned* | | | | |
| BART | 44.33 | 18.4 | 41.71 | 41.61 |
| VG-BART | 44.97 | 18.75 | 42.29 | 40.85 |
| BART-ImgCap | 44.91 | 18.54 | 42.12 | 41.34 |
| CMS-VG-BART (ours) | 45.13 | 18.81 | 42.56 | 42.13 |
| CMS-BART-ImgCap (ours) | **45.55** | **19.28** | **42.87** | **43.89** |
| T5 | 45.29 | 18.97 | 42.4 | 42.32 |
| VG-T5 | 45.58 | 18.94 | 42.75 | 42.3 |
| T5-ImgCap | 45.61 | 18.97 | 42.63 | 42.59 |
| LongT5 | 45.98 | 19.44 | 43.12 | 41.95 |
| LongT5-ImgCap | 46.6 | **19.86** | 43.7 | 42.63 |
| CMS-VG-T5 (ours) | 45.71 | 19.21 | 42.97 | 42.72 |
| CMS-T5-ImgCap (ours) | **47.29** | **19.86** | **44.13** | **44.74** |

Table 5: Results for the summarization task on mReddit-Sum. Models with "-ImgCap" in the name incorporate image information via image caption, and "VG-", via image embedding. Others are text-only models. Cluster-based multi-stage summarization (CMS) is our proposed method of processing discussions in three stages.

less than or equal to 22 turns. The performance on the long subset is noticeably worse than that on the short subset, lower by 4.95 ROUGE-1 and 6.1 BertScore.

To effectively handle this challenge, we present a novel method named **cluster-based multi-stage summarization (CMS)**, consisting of three stages (See Figure 1):

1. **Comment Clustering.** Similar comments are clustered using RoBERTa sentence embedding and agglomerative clustering.

2. **Cluster Summarization.** Each cluster is summarized in about a sentence using an LLM with image captioning, or a vision-guided LLM, such as VG-BART or VG-T5.

3. **Cluster-summary Summarization.** The cluster summaries are concatenated and further reduced into a coherent summary using a separate model, which is either an LLM with image captioning or a vision-guided LLM.

### 4.5 Implementation Details

The fine-tuned models are trained for 50 epochs on a single Titan RTX GPU for the BART and T5 models. We use a batch size of 4, and following (Yu et al., 2021; Raffel et al., 2020; Lewis

et al., 2020), we use learning rates 3e-5 to fine-tune the pre-trained parts of model weights, and 1.5e-4 to train the newly added visual layers in VG-BART and VG-T5. The decoding process uses beam-search with a size of 5. The average training time for BART, T5, BART-Cap, and T5-Cap is approximately 5 hours; the average training time for VG-BART and VG-T5 is about 8 hours, with the additional visual layers adding about 100 million extra parameters. We use the same training epochs, batch size, learning rates, and beam-search size for cluster-based multi-stage summarization. All results shown are an average of two runs.

## 5 Results and Analysis

Table 5 shows the results of all models evaluated across the test set. We see that our model, Cluster-based Multi-stage Summarization (CMS), outperformed baseline models for all metrics across both T5 and BART-based models. We believe this is due to our models' ability to better handle the long length of input threads, even outperforming LongT5 models; see § D.1 for more detailed analysis. In general across all model types, models that contain image information through an image caption outperform those that only have access to text-information. This supports that our dataset requires multimodal understanding in order to perform well on the summarization task. To confirm this, we additionally computed the CLIPScore between the image and the first sentence fo the generated summaries, which corresponds to the post summary and is where most of the image information is found. The results in Table 6 support that our methods incorporate more image information compared to a non-visual baseline. Vision-Guided models using text embeddings showed mixed results, with a marginal or no improvement over text-only models; we believe this to be due to a limitation of these models to effectively incorporate image information. Though they show strong performance on the How2 summarization task (Yu et al., 2021), mRedditSum has longer input and summary length, images, and fewer documents, likely contributing to the performance differences. Additionally, we note that T5 models show the best performance, followed by BART models and GPT3.5 models. For GPT3.5 models, we note that the low scores are likely due to inconsistencies in summary format, length, and detail, due to the zero-shot setting, but still receive relatively reasonable BertScore scores.

| Model | CLIPScore |
|---|---|
| BART | 68.29 |
| CMS-Bart-ImgCap | **70.26** |
| T5 | 69.34 |
| CMS-T5-ImgCap | **70.58** |

Table 6: The average CLIPScores computed between the first sentence of the generated summary and the threads' corresponding image.

We provide further analyses on the effect of input length and subreddit category on performance in § D.

### 5.1 Qualitative Analysis

In addition to our automatic evaluation, we check the test results manually for qualitative analysis. Several results can be found in Table 7. The primary advantage of our method, CMS, is that it has a greater coverage of relevant opinions compared to the baseline models. It is better able to filter out irrelevant or strange comments, while keeping the important opinions and including ones that are presented late in the thread.

We also find that all models, even those incorporating image information, are still prone to hallucinations of what is in the image. These include incorrect descriptions of object color and style, as well as describing objects that are not present in the image at all. Though our multimodal models are generally better at incorporating visual details than text-only models, their power to interpret the image seems still limited; we believe this to be due to potential undertraining of the text-vision fusion layers in the VG models, and the limitations of image caption models.

Thus, while our CMS model can overcome one weakness of the baseline multimodal summarization models, we still believe there to be significant room for improvement in the field of multimodal models, and hope that MREDDITSUM can help facilitate such research.

### 5.2 Human Evaluation

We additionally perform human evaluation studies via MTurk to compare the summaries generated from CMS-T5-ImgCap (ours) versus the baseline T5-ImgCap model. Based on similar works such as (Zhang et al., 2021), we use three metrics to measure the summary quality: fluency, faithfulness, and overall quality. **Fluency** measures which is

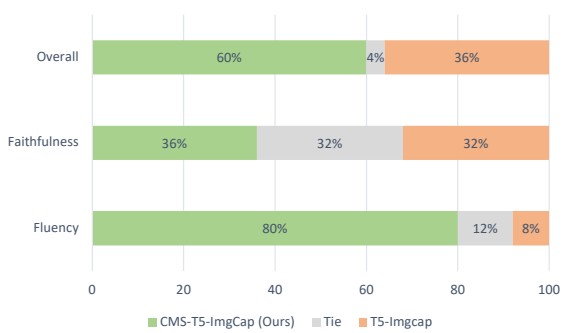

Figure 2: Human evaluation results of randomly sampled summaries of CMS-T5-Imgcap and T5-ImgCap models.

more naturally written, **faithfulness** measures how truthful the summary is to the document, and the **overall quality** represents general user preference. We randomly sample 25 datapoints from the test set and receive 3 annotations per sample. We note that this limited number of datapoints is due to the fact that this evaluation task is highly challenging for human annotators, given that the input, including the original post, threads, and image, is long and complex.

Figure 2 shows the majority vote results that our summaries are overall more preferable in terms of fluency and overall quality, with similar performance for faithfulness. We believe this to be an indicator of our model's better ability to effectively summarize the thread, offering more fluent and higher-quality summaries. The similar faithfulness scores are likely due to our method sharing the same base model and image caption as the baseline, T5-ImgCap, granting a similar ability to incorporate correct image and text information.

### 6 Conclusion and Future Work

Online discussions are increasingly becoming multimodal, yet there are not sufficient resources for summarizing them. To this end, we presented MREDDITSUM, the first multimodal discussion summarization dataset containing 3,033 discussion threads and images with human-written summaries. Threads were carefully chosen so that the images play a key role in the respective threads, and summaries were written to capture this. Experiments showed that summarization models making use of visual information consistently outperform those that do not. Additionally, we introduced Cluster-based Multi-stage Summarization, which accounted for the structure of discussion thread

| Image | Thread | T5 | T5-ImgCap | CMS-T5-ImgCap |
|---|---|---|---|---|
| | Original Post: Looking for ideas on how to style this necklace User 1: This gives me Ms. Frizzle vibes. Maybe embrace that? User 2: Yes! Funky skirt that poofs out, fitted shirt! User 3: Fitted button up collard white shirt with high waisted shorts and flats OP: Ooh that's a great idea! Thank you! User 4: [URL] Link to necklace if anyone is interested... | The OP asked for ideas on how to style a black and white floral necklace. One commenter suggested wearing the necklace with a fitted shirt. Another suggested wearing a button up collared white shirt with high waisted shorts and flats. | The OP wanted advice on how to style their necklace, which has a floral pattern on it. One commenter suggested wearing the necklace with a fitted shirt. Another suggested wearing a button up collared white shirt with high waisted shorts and flats. | The OP asked for ideas on how to style their necklace, which is white with a dinosaur pattern on it. One commenter suggested wearing a funky skirt and fitted shirt with the necklace. Another commenter recommended a button up collared white shirt with high waisted shorts and flats. One user shared a link to OP's necklace. |
| | Original Post: [Update] Couch arrived and this is the rug I ended up choosing. Any coffee table and accent chair recommendations? User 1: Something with a warm wood color would look nice. Check out antique shops or local furniture shops to find something that has some character and is made of solid wood. User 2: Yes, round or oval would look nice.... | The OP asked for advice on a coffee table and accent chair for their living room, which has a brown leather armchair and tan leather couch. Most commenters suggested a wood coffee table with a walnut finish and a solid white marble top. One commenter recommended a round or oval coffee table. Another commenter suggested brown throw pillows and blankets to match the rug. One user suggested OP get a non-shedding dog. | The OP asked for advice on a coffee table and accent chair for their blue couch. Most commenters suggested a wood coffee table with a walnut finish and a solid white marble top. One commenter recommended a round or oval coffee table. Another commenter suggested getting a non-shedding dog. | The OP asked for help with picking out a coffee table and accent chair for their blue couch. Most commenters suggested getting a warm wood coffee table. Others suggested a brown leather armchair or cream colored accent chair. One commenter suggested getting throw pillows and blankets to match the rug. Another commenter asked where the rug was from, and the OP said it was from Apt2B. |

Table 7: Examples of summaries generated from various models. Across all models, hallucinations regarding the image (highlighted in red) are present; however, these are reduced with multimodal models that incorporate image-only information (highlighted in green). Our CMS models tended to include more relevant details (blue) while removing irrelevant comments (orange).

data and outperformed baseline methods. We hope this dataset will help to facilitate active research on multimodal discussion summarization.

## Limitations

As with any dataset, there are some limitations to MREDDITSUM. Though it is of comparable size to many other summarization datasets, the relatively small size of the dataset makes it hard to utilize without significant pretraining, thus limiting the use of the dataset to those with access to large-scale pretraining datasets or pretrained models.

MREDDITSUM only includes Reddit threads with single images, as opposed to multiple images or videos. There is thus still room for improvement for multimodal summarization to additionally consider these threads.

Furthermore, our dataset considers only English-language threads from a single forum, Reddit, and a limited number of subreddits. There thus may be some additional bias due to the relatively small domain and raw nature of the dataset.

For our cluster-based multi-stage summarization method, one limitation is the need to train an extra model in addition to the base summarization model. As a result, our method incurs some computational overhead. However, it is worth noting that both the training and inference processes can be accommodated within a single Titan RTX GPU.

## Ethics Statement

As we propose a novel multimodal dataset, there are ethical considerations about the use of the data.

**Privacy**. All data are sourced from Reddit, which is publicly available. Following Desai et al. (2021); Ahn et al. (2023), additional measures have been taken to address privacy considerations. This includes the exclusion of images or discussions with clear identifying information, such as names or faces. Additionally, posts that are removed by their authors from Reddit also render the image unavailable for our dataset, as we only provide the links to the images. Thus, any users who are concerned about their post being in the dataset may easily remove it from the dataset by deleting it from Reddit.

**Bias**. As all data are sourced from real discussions on a public forum, there may be biases within the discussions due to the demographics of the Reddit users. Though we use a NSFW filter to remove inappropriate words, and look at each datapoint by hand to further filter out any harmful or inappropriate images or discussions, it is possible a few may still be present in the dataset. Less obvious bias such as stereotyping based on gender, etc. may also still be present in the dataset.

**Intended Use**. The MREDDITSUM dataset is intended to be used for research purposes only, and the use is subject to the the Reddit User Agreement, Privacy Policy, and Content Policy[3].

**Annotator Compensation**. We ensured that our annotators were paid a fair wage of approximately USD $16/hour, which is greater than the minimum wage in all countries that we recruited annotators from: The United States, Canada, Australia, New Zealand, and Great Britain. The time to complete each task was determined by running multiple trials with researchers, and the payment per task was calculated from this time. The cost per datapoint was approximately $3.50, with some longer datapoints costing more to compensate for the extra annotation time.

## Acknowledgements

This work was financially supported by SNU-NAVER Hyperscale AI Center, as well as the Institute of Information & Communications Technology Planning & Evaluation (IITP) grants funded by the Korea government (MSIT) (No. 2019-0-01082, SW StarLab and NO. 2021-0-01343, Artificial Intelligence Graduate School Program (Seoul National University)), and the National Research Foundation of Korea (NRF) grant funded by the Korea government (MSIT) (No. 2023R1A2C2005573). Joonsuk Park and Gunhee Kim are the corresponding authors.

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

## A  Subreddits Used

We list all subreddits used for data collection, along with the number of threads from each present in the final dataset in Table 8.

| Subreddit | Category | # Threads |
|---|---|---|
| r/outfits | Clothes | 161 |
| r/fashionadvice | Clothes | 529 |
| r/plussizefashion | Clothes | 19 |
| r/handbags | Clothes | 90 |
| r/petitefashionadvice | Clothes | 112 |
| r/weddingdress | Clothes | 108 |
| r/designmyroom | Interior | 1098 |
| r/malelivingspace | Interior | 642 |
| r/femalelivingspace | Interior | 258 |

Table 8: The subreddits used for data collection.

## B  Annotation Interface

We listed a total of 3 tasks on Amazon Mechanical Turk for our data pipeline. We informed all annotators that this data would be used to help in summarizing Reddit threads, asked them to agree with the Reddit Terms of Use before participating, and notified them that participating in the HIT constituted acceptance of these terms of use.

We provided annotators with detailed instructions for the task and several acceptable and unacceptable examples to help them perform the task. In Figure 6, we show the instructions provided for Task 1; similar instructions were used in the other two tasks. Additionally, we show the annotation interface used for Tasks 2 and 3 in Figure 7 and Figure 8.

## C  Additional Sample Data

We show a few additional data points from the MREDDITSUM dataset in Table 9 and Table 10. Table 9 shows a datapoint from the fashion category, whereas Table 10 shows a datapoint from the interior design category.

## D  Further Analyses

### D.1  Summarization based on the Length of Input Threads

To better understand whether CMS effectively handles long inputs, we run a further analysis using BART-based models (see Figure 3). As the number of comments increases, the R1 score consistently

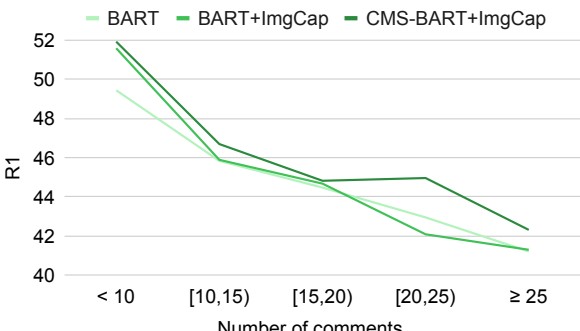

Figure 3: The influence of the number of comments in the thread on summarization performance (ROUGE-1) on BART-based models measured on the test set.

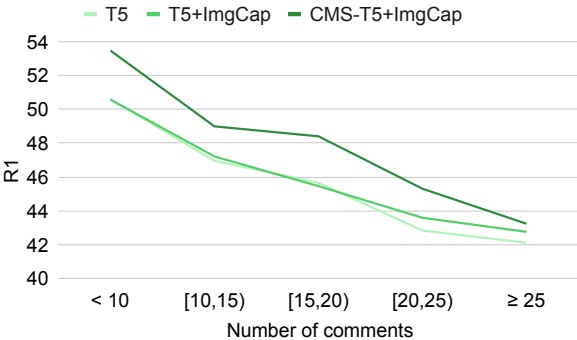

Figure 4: The influence of the number of comments in the thread on summarization performance (ROUGE-1) of T5-based models. The results are based on the test set.

decreases. This indicates that summarization indeed becomes more challenging when the input length is longer. However, the performance gap between the baseline models (i.e., BART, BART-ImgCap) and the CMS-BART-ImgCap generally increases as the number of comments grows, supporting the idea that CMS better handles longer threads. As our model generates cluster summaries in stage 1, it reduces the average input length by 82.8%, and thus achieving better performance even on relatively challenging long inputs. We also provide results from T5-based models in Figure 4, showing similar trends; the gap between the baseline models and the CMS-T5-ImgCap is large when the number of comments falls within the range of [15,20) and [20,25).

### D.2  Summarization per Subreddit

We further explore the summarization across 9 different subreddits, as shown in Figure 5.

The results reveal that subreddits within the 'Interior' category (i.e., the left three subreddits in Figure 5) exhibit lower ROUGE scores in compar-

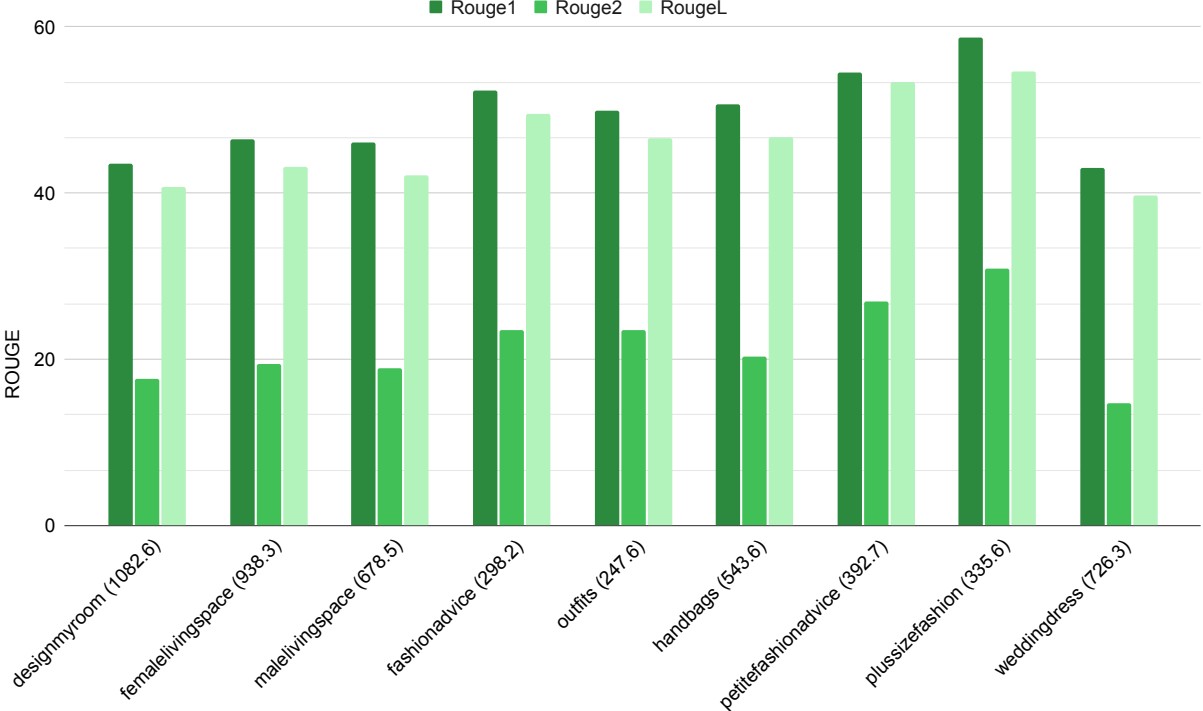

Figure 5: ROUGE scores obtained from our CMS-T5-ImgCap model on the test set, categorized by different subreddits. The number of input words is indicated in parentheses.

ison to subreddits within the 'Clothes' category (i.e., the right six subreddits in Figure 5). This discrepancy can be attributed to the difference in the input lengths across each subreddit. Given that the average input length of examples from the 'Interior' category exceeds that of examples from the 'Clothes' category, it is more difficult for our model to summarize the former. Additionally, we can also explain this gap by comparing the difference between domains. Specifically, while the model can easily comprehend clothing images by focusing on only salient objects, comprehending interior images is more challenging as it necessitates a broader range of information (e.g., wall color, spatial relationship between furniture, etc). Consequently, summarizing examples from the 'Interior' category proves to be more challenging for the models than summarizing examples from the 'Clothes' category.

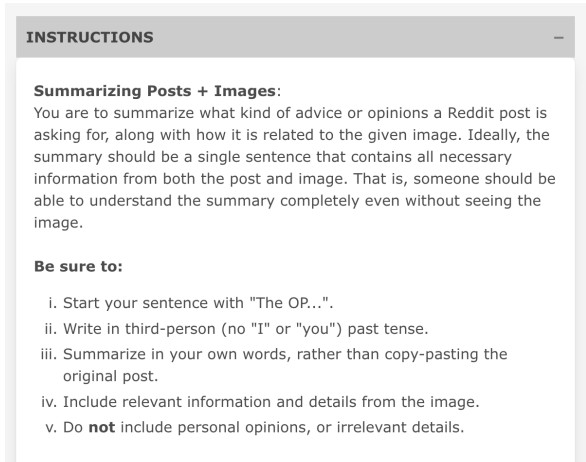

Figure 6: An example of instructions given for Task 1: Original Post Summarization.

**Post:**

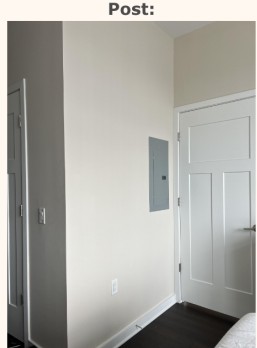

**OP:** *"what would you do to cover the ugly breaker box in my bedroom? obviously it still needs to be accessible, but i keep my door closed a lot and it's such an eyesore. the wall is 58 inches wide and 10 feet high."*

**Comments:**

Group 1:

- **User 4:** you could hang a large framed painting. something liked stretched canvas over frame. no glass. that way to can move it and not worry about glass breaking. some will even install a hinge on one side so it opens like a cabinet door.
- **User 6:** i think a painting or frame would be nice- you can hinge it if you want to give easier access. i feel like a gallery style wall with multiple pieces would be best otherwise your art would look off center and weirdly close to the corner, if that makes sense.
  - **OP:** makes total sense. the closeness to the edge of the wall is really what's throwing me off about it.
- **User 7:** hang something over it. a painting or some kind of rug.
- **User 14:** i would paint it the same color as the wall and hang 2-3 large frames on the wall. you could easily

◉ Write a summary sentence. You may write multiple sentences if necessary for this group.

> Enter your summary for group 1...

◯ No summary sentence is necessary - these comments are redundant or irrelevant.

Figure 7: An example of the Cluster Summarization task presented to workers on Amazon Mechanical Turk.

**Post 1:**

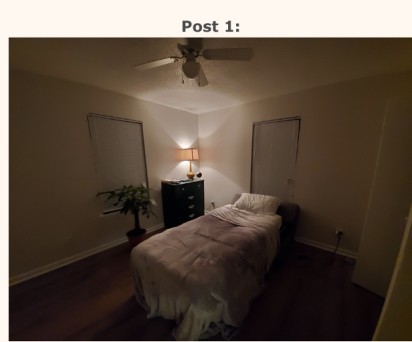

**OP:** *my room for about a year while flight instructing. what can i do to improve without being too permanent?*

**Original Summary 1:**

The OP wanted temporary solutions to improve their sparse bedroom with just a bed, dresser and fake plant. One user commented that the room looks like a dressing room for a rub-n-tug. One user recommended getting a rug. Another user suggested curtains which easy to put and add a lot of character and soften the place and a floor lamp. Another user suggested moving the bed so it is not directly under the window and adding a small nightstand to the room. The OP said the recommendation are great so far and is now planning. Another user suggested adding a life-size statue to make the place less lonely.

**Edited Summary 1:**

> The OP wanted temporary solutions to improve their sparse bedroom with just a bed, dresser and fake plant. One user commented that the room looks like a dressing room for a rub-n-tug. One user recommended getting a rug.
> Another user suggested curtains which easy to put and add a lot of character and soften the place and a floor lamp. Another user suggested moving the bed so it is not directly under the window and adding a small nightstand to the room. The OP said the recommendation are great so far

Figure 8: An example of the Summary Editing task presented to workers on Amazon Mechanical Turk.

**Image:**

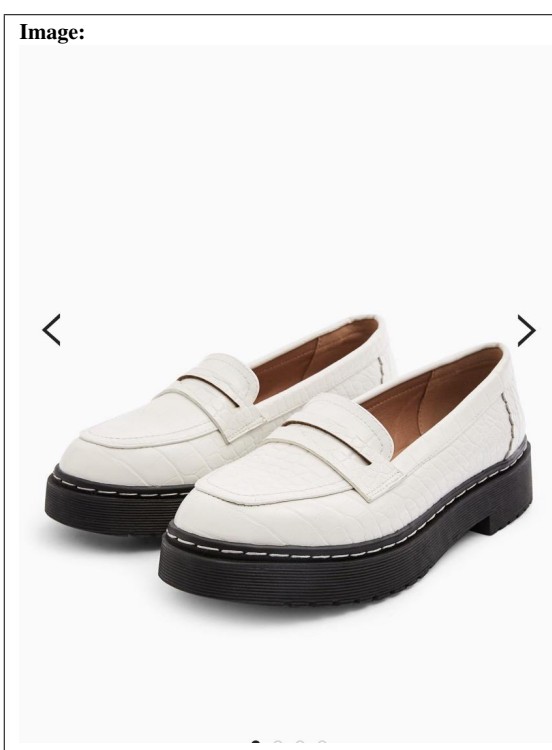

**Post Caption:**

what could you pair these with?

**Comments:**
**User 1:** Dressy black pants, colorful blouse, and blazer....

**User 2:** You can pair this with shorts, slacks, or jeans—basically anything. Just make sure that the color of your top & bottom matches.

**User 3:** If you are looking for women's wear I would say a very wide leg pastel high waisted pant with a tight/fitted top in same color scheme or white.

**User 4:** This reminds me of the kind of shoes I see in anime with sailor style uniforms tbh

**User 5:** A Goodwill donation

...

**Summary:**

The OP wanted to know what to wear with a pair of white leather loafers that have a thick black sole and low heel. One commenter thought pastel pants and a white top to match the shoes would work. Another commenter said that OP's shoes would pair with any sort of bottoms, but cautioned that the top and bottom color should match. One user shared links for OP to use as inspiration. Another user thought that the shoes looked like anime sailor shoes. Two commenters didn't like OP's shoes, and suggested they be thrown away or donated.

Table 9: Another example from our dataset, from the r/fashionadvice subreddit.

**Image:**

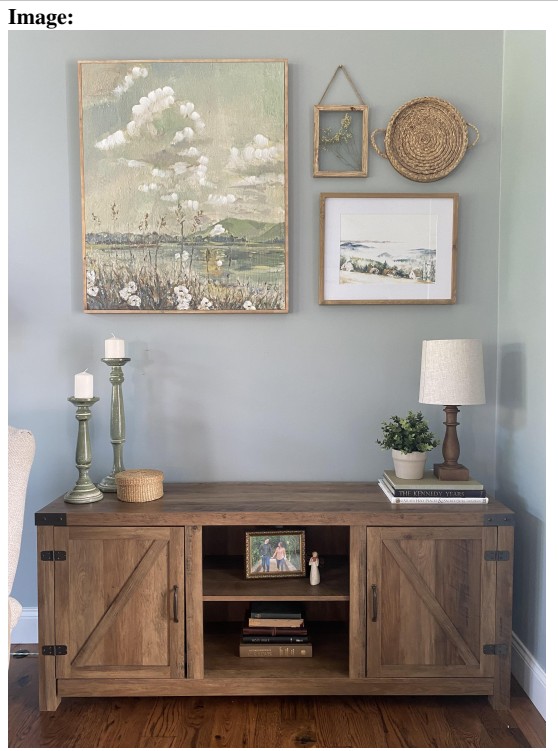

**Post Caption:**

Just moved into my first home and this space bothers me. Need some advice to make it look more cohesive.

**Comments:**
**User 1:** center feels empty. if it were me, i'd place one of those vintage wooden radio clocks in the middle. that's oddly specific i know...

**User 2:** Change nothing but add a vase of fresh white flowers in the center

**User 3:** Center large art piece and move it down. Lean the small art off center behind Candles, use the basket as a trinket tray on console.

**User 4:** I really like your art. I agree that the cneter needs something, maybe a plant or a stack of books.

**User 5:** I rather like it. The only thing missing is something sort of tallish in the center to fill that space. Like, it is the perfect spot for a vase filled with flowers. Some color and life! If a floral subscription isn't in your future lol maybe a full plant would fit the bill

...

**Summary:**

The OP asked what to do with a space in their home that presently has light blue walls and a brown sideboard with a lamp and candlesticks on it. Most commenters agreed the space looked good as-is, but recommended just adding something in the empty center of the table, such as a vase of white flowers or a large plant. Others thought a vintage wooden radio clock or traditional record player in the same green color as the candles would look perfect, while another suggested a stack of nice books. Others said to center the large wall art, and to check local thrift stores for a substantial but short statement piece to be the center accent decor. Others recommended using a basket as a trinket tray or just buying a marble tray for trinkets on the table. They also said to lean the smallest art pieces against the wall behind the candles, or get rid of the candles altogether.

Table 10: Another example from our dataset, from the r/designmyroom subreddit.