# OpenReview forum: "mRedditSum: A Multimodal Abstractive Summarization Dataset of Reddit Threads with Images"
_EMNLP/2023/Conference — EMNLP 2023 Main_

### Official Review · Reviewer_BJQs · 2023-07-21

**Soundness:** 4

**Excitement:**

3: Ambivalent: It has merits (e.g., it reports state-of-the-art results, the idea is nice), but there are key weaknesses (e.g., it describes incremental work), and it can significantly benefit from another round of revision. However, I won't object to accepting it if my co-reviewers champion it.

**Paper Topic And Main Contributions:**

The paper presents a new dataset for multimodal discussion summarization that is curated from Reddit. The task is to create an abstractive summary that takes into account the original post text, attached image and comments from other users. The dataset creation was done by employing annotators. The next contribution of the paper is the CMS summarization architecture that works by clustering the content in multiple steps and resulting in a single summary. The new dataset has been evaluated on 3 other text models including the proposed CMS model. Since the task is multimodal, 2 different model architectures have been chosen: image captioning output and the task text, fine-tuning textual and visual encoders (VG-BART or VG-T5) on the task. Based on the results presented in Table 5, CMS-T5-ImgCap the model obtains the highest score among compared variants of other models.

**Questions For The Authors:**

Is not possible to compare the models across other datasets that are listed in Table 2?

**Reasons To Accept:**

The proposed dataset and the CMS model are the main contributions of the papers. Both are well-described and written with clear details.

**Reasons To Reject:**

The only criticism is the lack of comparability of CMS on other datasets. Showing its performance on a single dataset (the proposed dataset) is somewhat limited.

**Reproducibility:**

4: Could mostly reproduce the results, but there may be some variation because of sample variance or minor variations in their interpretation of the protocol or method.

**Reviewer Confidence:**

3: Pretty sure, but there's a chance I missed something. Although I have a good feel for this area in general, I did not carefully check the paper's details, e.g., the math, experimental design, or novelty.

---

> ### Author Rebuttal · Authors · 2023-08-28
>
> We are glad that the reviewer found the contributions easy to understand and well-described, and appreciate that the reviewer clearly understood the main points of the paper and wrote a thorough summary. We would like to respond to the reviewer’s main concern below.
>
> **"The only criticism is the lack of comparability of CMS on other datasets…is it not possible to compare it across other datasets listed in Table 2?"**
>
> Our main research question of this work is the summarization of multimodal posts with a long thread of discussion containing multiple, relevant viewpoints from different users. Thus, the input document must be naturally split into clusters, or groups of relevant comments/topics, and include summaries of these clusters. Unfortunately there are few summarization datasets that meet this criteria - the closest one within Table 2 is AnswerSumm, which does include clustered groups of comments, cluster summaries, and a final summary. Given that AnswerSumm is text-only, we originally felt that CMS may not be appropriate for AnswerSumm due to its lack of multimodal information, which is a key part of our experiments. Additionally, AnswerSumm has a far shorter number of comments (average 6.4 vs mRedditSum’s 22.6, as shown in Table 2), potentially weakening the benefit that CMS provides from filtering out long and irrelevant comments. Finally, though AnswerSumm has cluster and cluster summaries, these are human-selected and do not use the agglomerative clustering algorithm that we did for mRedditSum and CMS, which means the final summarization pipeline cannot be fully automated.
> Despite these limitations, we decided to attempt CMS on this dataset as well, using the human-annotated cluster labels provided by AnswerSumm. We found that this technique significantly improved scores, beating the baseline by a total of 6.75 Rouge-2 points for T5 based models. It should be noted that as this method uses the human-annotated clusters during test time, it is not fully comparable to the baseline results that do not use this information. However, we believe it shows the effectiveness of CMS when clustering information is available.
>
> |                    | R-1   | R-2   | R-3   | BERTScore |
> |--------------------|-------|-------|-------|-----------|
> | T5                 | 26.2  | 9.38  | 23.27 | 25.28     |
> | CMS-T5-GoldCluster | 33.98 | 16.13 | 31.0  | 35.24     |
>
> Thank you again for the kind review - we appreciate the reviewer’s suggestion, and hope that our response brought some further clarity to this topic.

---

### Official Review · Reviewer_2UFg · 2023-08-03

**Soundness:** 4

**Excitement:**

3: Ambivalent: It has merits (e.g., it reports state-of-the-art results, the idea is nice), but there are key weaknesses (e.g., it describes incremental work), and it can significantly benefit from another round of revision. However, I won't object to accepting it if my co-reviewers champion it.

**Paper Topic And Main Contributions:**

The paper introduces a multimodal summarization dataset based on Reddit threads along with the comments. The authors have annotated summaries along with a pipeline bench mark approach. Evaluations indicate that the proposed approach is competetive to summarize threads and also incorporates details from the images.

**Reasons To Accept:**

A dataset of this nature will be valuable to the community.

**Reasons To Reject:**

1. Looking at the length of the posts, it is not clear if this is really a summarization task.
2. The proposed approach's ability to capture multimodal information is largely anecdotal without any extensive evaluation of the same.

**Reproducibility:**

3: Could reproduce the results with some difficulty. The settings of parameters are underspecified or subjectively determined; the training/evaluation data are not widely available.

**Reviewer Confidence:**

4: Quite sure. I tried to check the important points carefully. It's unlikely, though conceivable, that I missed something that should affect my ratings.

---

> ### Author Rebuttal · Authors · 2023-08-28
>
> We appreciate that the reviewer noted the value of the dataset to the community and summarized the key contributions well. We would like to address the reviewer’s concerns below.
>
> **"Looking at the length of the posts, it is not clear if this is really a summarization task."**
>
> We would first like to note that our summarization results in this work focus on the Full Thread summary, which as described in Sec 3.3., is on average 13.2% as long as the input (comparable to SamSUM’s 19% and How2’s 11.3%). We believe this reduction in word length is a clear indication of a summarization task.
> We believe that the review may be referring to the Original Post summary described in Sec 3.2. and Table 3, where on average the document is 18.87 words while the summary is 23.14 words. Although the summary is on average longer for this structure, the key thing to note is that this is a multimodal summary - that is, we are also summarizing an image along with the text. When the text is short and the image is crucial for understanding - as in the Original Post section of a thread - it is not unexpected that describing the image would add a small amount of length to the summary. We believe multimodal summarization may not strictly adhere to the idea that the summary must be fewer words than the text component of the document, as it is condensing image information into the summary as well.
>
> **"The proposed approach’s ability to capture multimodal information is largely anecdotal without any extensive evidence of the same."**
>
> As the main focus of our CMS summarization technique was to better capture multiple, relevant viewpoints from the thread, we highlighted the improvements in the overall ROUGE and BERTScore scores in our paper’s analysis. As can be seen in Table 5 and is discussed in Sec. 5, models that contained image information through an image caption showed improvements across all scores, across all model types. As the only difference in the model input across these models was the multimodal information captured by the image caption, we believe the increase in scores can only be attributed to a better capture of the necessary multimodal information via the automatically generated image caption.
> However, we agree that a more thorough comparison focusing solely on image information could be useful. To address this, we additionally computed the average CLIPScore between the image and the first sentence of the generated summaries, which corresponds to the post summary and is where most of the image information is found. The scores are as following:
>
> | Model           | CLIPScore |
> |-----------------|-----------|
> | T5              | 69.34     |
> | CMS-T5-Imgcap   | **70.58**     |
> | BART            | 68.29     |
> | CMS-BART-Imgcap | **70.26**     |
>
>
> These scores provide further evidence that our proposed method significantly increases the amount of multimodal information provided in the summary through the use of an automatically generated image caption across both T5 and BART based models, as expected. We will include this additional analysis in the paper.
>
> We thank you again for the kind review, and hope that our feedback provided some clarity on the reviewer’s concerns.

---

### Official Review · Reviewer_anng · 2023-08-04

**Typos Grammar Style And Presentation Improvements:** The paper is, in general, readable
**Soundness:** 4

**Excitement:**

3: Ambivalent: It has merits (e.g., it reports state-of-the-art results, the idea is nice), but there are key weaknesses (e.g., it describes incremental work), and it can significantly benefit from another round of revision. However, I won't object to accepting it if my co-reviewers champion it.

**Paper Topic And Main Contributions:**

The paper presents a new multimodal abstractive summarization dataset (MRedditSum), which contains 3,033 Reddit threads (post+comments) with text (posts and comments) and images (posts). For each thread, a human-written summary was collected first for the post (text and image) and then for the top-5 comment clusters. The post summary and comments summaries were later merged by a human annotator to generate a full summary. Additionally, the paper proposes a new cluster-based multi-stage summarization method that outperforms baselines (text-only, text with the caption, and text with image) on this new dataset. The experiments also show that the performance of current summarization models improves by including visual information.

**Questions For The Authors:**

A. Figure 2 shows the majority vote results among three human annotators, as explained in Section 5.2. Please explain the significance of the tie.

B. Figure 2's results show very high fluency for the proposed model compared to the baseline, even though the language model remains the same (T5). However, this result does not correlate with Table 6. On the other hand, the faithfulness score is similar even when the proposed model prunes comments using clustering.

**Reasons To Accept:**

A new multimodal abstractive summarization dataset based on Reddit discussions, consisting of 3033 threads and human-written summaries.

The paper proposes a new method for multimodal summarization on this new dataset, which outperforms the existing baselines.

The proposed dataset is novel and has the potential to contribute to the field of multimodal abstractive summarization.

**Reasons To Reject:**

Lack of detail on the total number of annotators and whether the same annotator generated the comment and full summary for the study. Additionally, the paper does not provide an inter-annotator agreement for a few samples to understand the variance in the quality of the summaries across the dataset.

Comparison with a text-only baseline model that supports longer sequence lengths, such as LongFormer/LongT5, is missing. The input sequence length of the document (post+full thread) exceeds 512 tokens. It will also strengthen the claim of the proposed model's ability to handle longer input lengths as discussed in Section 5.

**Reproducibility:**

3: Could reproduce the results with some difficulty. The settings of parameters are underspecified or subjectively determined; the training/evaluation data are not widely available.

**Reviewer Confidence:**

3: Pretty sure, but there's a chance I missed something. Although I have a good feel for this area in general, I did not carefully check the paper's details, e.g., the math, experimental design, or novelty.

---

> ### Author Rebuttal · Authors · 2023-08-28
>
> We appreciate that the reviewer took the time to understand the paper well, and correctly identify our main contributions of introducing a novel multimodal dataset and a new summarization method that have the potential to contribute to the field of multimodal abstractive summarization.
>
> **"Lack of detail on the total number of annotators and whether the same annotator generated the comment and full summary for the study."**
>
> Summaries were written by a pool of 40 annotators who passed our qualification tests. Since each summarization stage was an independent task on AMT and annotators were randomly assigned a thread to summarize, the annotators differed from one stage to another for a given thread. We felt that as all annotators passed strict qualification tests, this did not lead to any degeneration in annotation quality. We will clarify this in the paper.
>
> **"…The paper does not provide an inter-annotator agreement for a few samples to understand the variance in quality of the summaries."**
>
> Unlike that of categorical labels, the quality of written summaries is difficult to quantify with inter-annotator agreement scores, as the notion of “agreement” is not well defined. Thus, rather than measuring the summary qualities with such a metric, we carefully controlled the quality of annotators through a series of filtering procedures. As described in Sec. 3.2, all annotators came from English-speaking countries, had a HIT approval rate over 98%, more than 5000 HITs approved, and passed a strict qualification test where each result was manually checked for quality.
> However, we agree that measuring overall summary quality can be beneficial.
> Following a closely related work, SamSUM (Gliwa et al., 2019), we randomly selected 100 thread-summary pairs and had 2 independent judges grade them on a quality scale of -1, 0, or 1, where -1 means a summary is irrelevant or does not make sense, 0 means a summary extracts only part of the relevant information or makes some mistakes, and 1 means a summary that is understandable and delivers a brief overview of the relevant information from the thread. We asked annotators to score summaries on overall quality, fluency, and faithfulness, similar to our human annotation found in Sec. 5.2. We found the scores were highly positive, with average scores of 0.95, 0.96, 0.83 for overall quality, fluency, and faithfulness respectively. Additionally, we found Gwet's AC1 agreement scores of .91, 0.89, and 0.53, corresponding to high, high, and moderate agreement, respectively. Note that we used the Gwet's AC1 score for interannotator agreement as it performs well despite class imbalances where agreement is high. (Gwet 2008, Wongpakaran et al. 2013, Wong et al. 2021).
>
>
> **"Comparison with a text-only baseline model that supports longer sequence lengths, such as LongFormer/LongT5, is missing."**
>
> We agree that the addition of one of these models could provide an interesting comparison, and strengthen the claim that the proposed model is better able to handle longer inputs. As LongFormer is a BERT-like model incapable of text generation alone, we chose to experiment with the LongT5, which also provides a better comparison with our best model. We thus pretrained a LongT5-base model on CNN/DailyMail, as described in Sec. 4.3, and fine-tuned both a text-only and an image-caption based extension. We found the following results:
>
> |                      | R-1   | R-2   | R-3   | BERTScore |
> |----------------------|-------|-------|-------|-----------|
> | LongT5               | 45.98 | 19.44 | 43.12 | 41.95     |
> | LongT5-ImgCap        | 46.6  | **19.86** | 43.7  | 42.63     |
> | CMS-T5-ImgCap (ours) | **47.29** | **19.86** | **44.13** | **44.74**     |
>
> These results show that while LongT5 outperforms regular T5 in all metrics except BERTScore, our CMS method still outperforms LongT5 despite LongT5’s additional attention mechanism and longer input handling.
>
> **"Please explain the significance of the tie in Figure 2."**
>
> For the human evaluation experiments, annotators were also given the option to choose a 'Tie' between the two summaries presented to them. We allowed this to create a more fair comparison between model results, without forcing annotators to choose a better model in case they did not notice a significant difference between the summaries. While we focused on the percentage of Wins/Losses for analysis in the paper, the low percentage of ties for the Overall and Fluency scores can be interpreted as a stronger preference, while the higher percentage of ties for Faithfulness show a lower preference between summaries, which is consistent with the percentage of Wins/Losses.
>
> **"Figure 2 shows high fluency and similar faithfulness for proposed model compared to baseline, despite using the same language model and pruned comments. This does not correlate with Table 6."**
>
> We believe this to be an expected result, and argue that it does in fact correlate with the example summaries presented in Table 6. Although we understand fluency may have differing definitions, we defined fluency as how naturally written the summary is, described in Sec 5.2. We believe that as the proposed model more naturally captured the varying viewpoints of the commenters, without including irrelevant comments, annotators likely felt these were more naturally written. This can be seen in the second example of Table 6, where the proposed model captures more relevant details on the accent chair suggestions without discussing the irrelevant non-shedding dog. Even though the language generation ability of the model was the same and the readability of each summary was similar, the improved content of the summary likely impacted its fluency score as annotators may feel this is more ‘natural’.
> Additionally, faithfulness was defined as how truthful the summary is to the document. The faithfulness score remains similar between models as both methods utilize the information from the image caption as well as the comments, and thus both have a similar hallucination rate and hence a similar faithfulness score; this can be seen in Table 6 where both the proposed method and baseline ImgCap method either had a hallucination as in the first example, or no hallucination in the 2nd example. We may see lower faithfulness scores with models that did not have image information available at all.
>
> Thank you again for the kind feedback - we appreciate the detailed comments and hope we have addressed most concerns!
>
> **References**
>
> [1] Gliwa et al., "SAMSum Corpus: A Human-annotated Dialogue Dataset for Abstractive Summarization", ACL workshop 2019.
>
> [2] Gwet, "Computing inter-rater reliability and its variance in the presence of high agreement", British Journal of Mathematical and Statistical Psychology 2008.
>
> [3] Wongpakaran et al., "A comparison of Cohen’s Kappa and Gwet’s AC1 when calculating inter-rater reliability coefficients: a study conducted with personality disorder samples", BMC Medical Research Methodology 2013.
>
> [4] Wong et al., "Cross-replication Reliability - An Empirical Approach to Interpreting Inter-rater Reliability", ACL 2021.

---

### Meta-Review · Area_Chair_4KJb · 2023-09-18

**Recommendation:** 4

**Metareview:**

This paper presents mRedditSum, a new multimodal abstractive summarization dataset, and a novel cluster-based multi-stage summarization method. The dataset consists of 3,033 Reddit threads where a post solicits advice regarding an issue described with an image and text, and respective comments express diverse opinions. These threads are annotated with a human-written summary that captures both the text's essential information and the image's details. The paper also shows that the performance of summarization models, specifically the proposed CMS model, improves when visual information is incorporated.

Reviewers acknowledge the novelty and potential contribution of the mRedditSum dataset and the proposed cluster-based multi-stage summarization method to the field of multimodal abstractive summarization. The paper is well-written, with clear details about the proposed dataset and method. However, there are also concerns that may merit attention. These include a lack of detail on the total number of annotators and whether the same annotator generated the comment and full summary for the study. It is also pointed out the absence of an inter-annotator agreement for a few samples and a comparison with a text-only baseline model that supports longer sequence lengths.

Overall, this paper proposes a potentially useful dataset for the research community, as well as a summarization model that surpasses SOTA. Authors are encouraged to address the reviewers' comments, in particular the concerns on the human annotation process, and their rebuttals into the final version if accepted.

---

### Decision · Program_Chairs · 2023-10-07

**Decision:**

Accept-Main

**Comment:**

This paper presents mRedditSum, a new multimodal abstractive summarization dataset, and a novel cluster-based multi-stage summarization method. The dataset consists of 3,033 Reddit threads where a post solicits advice regarding an issue described with an image and text, and respective comments express diverse opinions. These threads are annotated with a human-written summary that captures both the text's essential information and the image's details. The paper also shows that the performance of summarization models, specifically the proposed CMS model, improves when visual information is incorporated.

Reviewers acknowledge the novelty and potential contribution of the mRedditSum dataset and the proposed cluster-based multi-stage summarization method to the field of multimodal abstractive summarization. The paper is well-written, with clear details about the proposed dataset and method. However, there are also concerns that may merit attention. These include a lack of detail on the total number of annotators and whether the same annotator generated the comment and full summary for the study. It is also pointed out the absence of an inter-annotator agreement for a few samples and a comparison with a text-only baseline model that supports longer sequence lengths.

Overall, this paper proposes a potentially useful dataset for the research community, as well as a summarization model that surpasses SOTA. Authors are encouraged to address the reviewers' comments, in particular the concerns on the human annotation process, and their rebuttals into the final version if accepted.